# Quality Factor Improvement of a Thin-Film Piezoelectric-on-Silicon Resonator Using a Radial Alternating Material Phononic Crystal

**DOI:** 10.3390/mi14122241

**Published:** 2023-12-15

**Authors:** Chuang Zhu, Muxiang Su, Temesgen Bailie Workie, Panliang Tang, Changyu Ye, Jing-Fu Bao

**Affiliations:** School of Integrated Circuit Science and Engineering, University of Electronic Science and Technology of China, Chengdu 611731, China; zc20082012@163.com (C.Z.); 202322310701@std.uestc.edu.cn (M.S.); wtbailie@std.uestc.edu.cn (T.B.W.); mailfortpl@163.com (P.T.); yechangyu@std.uestc.edu.cn (C.Y.)

**Keywords:** RAM-PnC, resonator, anchor loss

## Abstract

This paper studies the radial alternating material phononic crystal (RAM-PnC). By simulating the band gap structure of the phononic crystal, a complete acoustic band gap was verified at the resonant frequency of 175.14 MHz, which can prevent the propagation of elastic waves in a specific direction. The proposed alternately arranged radial phononic crystal structure is applied to the thin-film piezoelectric-on-silicon (TPOS) MEMS resonator. The finite element simulation method increases the anchor quality factor (*Q_anchor_*) from 60,596 to 659,536,011 at the operating frequency of 175.14 MHz, which is about 10,000 times higher. The motion resistance of the RAM-PnC resonator is reduced from 156.25 Ω to 48.31 Ω compared with the traditional resonator. At the same time, the insertion loss of the RAM-PnC resonator is reduced by 1.1 dB compared with the traditional resonator.

## 1. Introduction

The radiofrequency microelectromechanical system (RF-MEMS) is an essential branch of MEMS technology. The advantages of RF-MEMS devices, such as low power consumption, miniaturization, and integration with CMOS (Complementary Metal Oxide Semiconductor) circuits, are urgently needed in future RF communication systems. With the advent of the 5G era, a mobile phone needs to meet all the communication bands of 3G, 4G, and 5G at the same time, as well as Bluetooth, Wi-Fi, and GPS, so the number of filters in the mobile phone terminal is greatly increased to about 100. Therefore, various RF frontends have an urgent need for filters with small sizes, low power consumption, and high performance [1]. As the core of the filter, the size, power consumption, and performance of the resonator have an essential influence on the index of the filter. Many researchers have conducted a lot of work to improve its performance and quality factor [2,3,4]. 

In 2011, Harrington and Abdolvand [5] introduced an acoustic mirror structure that could reflect the elastic waves propagating from the resonator, and the *Q* enhancement was as high as 560%. In 2016, Tu and Lee [6] changed the displacement distribution on the resonant body by optimizing the structure of the resonant body itself to minimize the vibration at the support beam, thereby reducing the acoustic wave dissipated outward through the support beam. In 2017, Tu and Lee [7] proposed a stress-release hole structure, and this hole redistributed the strain energy in the resonant cavity. The redistribution of strain energy significantly inhibits the axial deformation of the support beam anchor cable and destroys the anchorage zone. Reducing the anchor cable’s energy and the anchorage area can reduce anchorage loss and improve *Q*.

In 2017, Zou et al. [8] proposed a butterfly aluminum nitride (AlN) plate with an anchor plate angle of less than 90°. By eliminating the anchor point loss and improving the quality factor (*Q*) of the LWR resonator, the circular butterfly plate had a better suppression effect on the anchor loss than the inclined butterfly plate. In addition, the emerging phononic crystal theory and technology in recent years have greatly improved the performance of MEMS resonators. In previous studies, many types of phononic crystals have been proposed, such as hole circles, air holes [9], rings [10], solid disks [11], cross-shape [12], spider web-like shapes [13], and so on [14,15,16,17,18,19,20,21,22,23,24,25,26,27]. However, it is relatively rare to use radial phononic crystals to improve the quality factor of resonators. Therefore, this paper discusses the effect of radial phononic crystals on the performance of MEMS resonators.

The theoretical analysis of phononic crystals (PnC) uses a novel class of periodic synthetic materials to manipulate the propagation of elastic waves and acoustic waves [28]. Based on traditional phononic crystals, some researchers have proposed radial phononic crystal structures, which are arranged periodically in the radial direction and have a huge advantage under certain circumstances. Torrent et al. [29,30,31] studied the propagation characteristics of bulk waves in radial phononic crystal and found that the propagation of radial bulk waves in a specific frequency band was prohibited. Li et al. [32] and Ma et al. [33] constructed two types of radial phononic crystals and studied the propagation characteristics of Lamb waves in them. In 2015, Shu et al. [34] proposed a generalized one-dimensional cylindrical shell (CS GPCs) phononic crystal. In this structure, two homogeneous materials were periodically arranged along the radial direction. The CS GPCs possess radial, torsional shear, and axial shear wave band gaps in the high-frequency region, which comply with the Bragg scattering effect. In 2016, Shi et al. [35] proposed that two alternate homogeneous materials could be periodically introduced along the radial direction to form a circular plate of radial phononic crystals (CPRPC). Though radially periodic, there are significant transverse and longitudinal wave band gaps. In 2018, Li et al. [32] proposed a composite radial plate-type elastic metamaterial composed of periodic double-sided composite branches on a one-dimensional binary radial phononic crystal plate. In 2018, An et al. [36] proposed a two-dimensional cylindrical shell structure with radial and circumferential periods. Through the calculation and discussion of wave transmission characteristics, it was found that the radial wave in the design showed apparent attenuation in a particular frequency region, that is, the wave band gap.

Based on the work and research of the above researchers, this paper first proposes the RAM-PnC structure. Combining Si and metal W into a multi-material design has a wide band gap range and avoids the strength damage of the etched substrate to the resonator anchor point. Then, the proposed structure’s dispersion and transmission curves are calculated, and the effects of different materials and structural parameters on the band gap are discussed. Finally, it is applied to the TPOS MEMS resonator designed in this paper, significantly improving its performance.

## 2. Phononic Crystal and the Theory of Wave Propagation

### 2.1. Phononic Crystal Structure and Band Gap Calculation

As shown in Figure 1, we constructed a three-dimensional schematic diagram of a one-dimensional radial phononic crystal composed of two alternating materials (green represents W, gray represents Si). In Figure 1, the radial phononic crystal cylindrical shell is a one-dimensional cylindrical semi-infinite periodic structure formed by two kinds of solid dielectric materials, W and Si, with different elastic constants and densities in the r direction. The width a_1_ of the W layer is 6 μm, the width a_2_ of the Si layer is 10 μm, and the height of the W layer and the Si layer is H = 10 μm. A radial periodic unit is alternately formed by W and Si, and its length is a = 16 μm. Silicon is an anisotropic material, and its crystal orientation and elastic coefficient influence the simulation results. In this paper, the parameters of Si are the crystal orientation, and the elastic coefficients are shown in Table 1.
(1)$$\rho \frac{{𝜕}^{2}u}{𝜕t^{2}}=\left(\lambda +2\mu \right)\frac{𝜕\theta_{t}}{𝜕r}-\frac{2\mu }{r}\frac{𝜕w_{z}^{\prime }}{𝜕\theta }+2\mu \frac{𝜕w_{\theta }^{\prime }}{𝜕\theta }$$
(2)$$\rho \frac{{𝜕}^{2}v}{𝜕t^{2}}=\left(\lambda +2\mu \right)\frac{𝜕\theta_{t}}{𝜕r}-2\mu \frac{𝜕w_{z}^{\prime }}{𝜕\theta }+2\mu \frac{𝜕w_{r}^{\prime }}{𝜕r}$$
(3)$$\rho \frac{{𝜕}^{2}w}{𝜕t^{2}}=\left(\lambda +2\mu \right)\frac{𝜕\theta_{t}}{𝜕z}-\frac{2\mu }{r}\frac{𝜕w_{z}^{\prime }}{𝜕\theta }+\frac{2\mu }{r}\frac{𝜕w_{r}^{\prime }}{𝜕\theta }$$
where *u*, *v*, and *w* are displacements, *ρ* is the mass density, *t* is time, *λ* and *μ* are Lame constants, and *r*, *θ* and *z* represent coordinate variables in cylindrical coordinates, respectively. In addition, the volume strain and the rotational component are defined through the following formulas.
(4)$$\theta_{t}=\frac{1}{r}\frac{𝜕\left(ru\right)}{𝜕r}+\frac{1}{r}\frac{𝜕u}{𝜕\theta }\frac{𝜕w}{𝜕r}$$
(5)$$w_{r}^{\prime }=\frac{1}{2}\left(\frac{1}{r}\frac{𝜕w}{𝜕\theta }+\frac{1}{r}\frac{𝜕v}{𝜕z}\right)$$
(6)$$w_{\theta }^{\prime }=\frac{1}{2}\left(\frac{1}{r}\frac{𝜕u}{𝜕\theta }+\frac{𝜕w}{𝜕r}\right)$$
(7)$$w_{z}^{\prime }=\frac{1}{2}\left(\frac{1}{r}\frac{𝜕\left(rv\right)}{𝜕\theta }+\frac{𝜕u}{𝜕\theta }\right)$$

COMSOL multiphysics solved the intrinsic Equations (1)–(4). In the stress–strain application model, the displacement *v* and the stress and strain components in the direction are assumed to be 0. In this model, the displacements of *u_or_* and *z* are defined. The dependent variable *u_or_* ≡ *u*/*r* is introduced to avoid dividing *r*, leading to axis problems. In Equations (4)–(7), *r* = 0, *w* is the displacement in the *z* direction. Periodic boundary conditions are imposed on the element in the *r* direction as follows:(8)$$\mu \left(r+a,z\right)=\mu \left(r,z\right)e^{ik_{r}a}$$
where *r* is the radial position, a is the lattice constant, and the parameter *k_r_* is defined as a one-dimensional block wave vector along the radial direction. The free boundary is applied to the plate surface along the *z* direction. By scanning the wave vector *k_r_* along the boundary of the first Brillouin zone, the dispersion curve ω = ω (k) and the eigen-displacement field can be obtained. The structure used in this section is one-dimensional in the radial direction, so the Brillouin zone boundary ranges from Γ (0,0) to R (1,0), which is different from the two-dimensional or three-dimensional phononic crystal structure.

### 2.2. Band Gap Optimization of Phononic Crystals

Different materials affect the band gap of the structure. Mass density and Young’s modulus are two main factors in adjusting the band gap for solid–solid phononic crystals. The damping materials can include the following six common metals: W, Al, Ag, Pt, Cu, and Au. Detailed material parameters are shown in Table 2. By comparing the acoustic parameters in the table, we found that the acoustic impedance of metal W is the largest. Furthermore, we simulated the dispersion curves of different materials in the frequency range of 0–280 MHz. Figure 2a shows the wide band gap of 122–225 MHz. Figure 2b is Al metal, and its band gap range is 198–218 MHz. Figure 2c represents the Ag metal, and its band gap ranges are 165–171 MHz, 231–237 MHz, and 238–242 MHz. Figure 2d represents the Pt metal, and its band gap ranges are 110–141.5 MHz, 156–170 MHz, 172–183 MHz, 228.5–238.5 MHz, and 247–263 MHz. Figure 2e represents the Cu metal with a band gap range of 158–177 MHz. Figure 2f represents the Au metal, and its band gap range is 120–136.5 MHz, 168–175 MHz, 192–208 MHz, 227–234 MHz, 250–255 MHz, 262–272 MHz, 273.5–277.5 MHz. By comparing the band gap range generated by the combination of the above six metals and Si, the combination of W and Si was shown to produce the best effect, with a wider band gap width and, thus, an excellent acoustic isolation effect.

The band gap range is further analyzed when other parameters are kept unchanged, and only the metal width w is changed. The calculation results are shown in Figure 3. It can be seen from the figure that when the metal width w gradually increases from 2 μm to 30 μm, the initial band gap and the final band gap of the first band gap decrease, and the band gap becomes wider and narrower. When the metal width w = 6 μm, the widest band gap of 122–225 MHz is obtained.

### 2.3. Transmission Characteristic

By calculating the elastic wave transmission curve, the existence and ability of the proposed RAM-PnC acoustic band gap can be evaluated. Different from the infinite scattering array used for the dispersion relation, the model used for transmission spectrum analysis is a finite scattering array. As shown in Figure 4, there are five periodic units arranged alternately in the radius direction. A is the excitation port, B is the receiving port, and the outermost layer is the PML layer with a length of 5a. The transmission loss is defined by the following equation:(9)$$S_{21}=10log_{10}\left(\frac{P_{out}}{P_{in}}\right)=10log_{10}\left(\frac{d_{out}^{2}}{d_{in}^{2}}\right)$$

The harmonic excitation of the radial displacement in the direction of r is applied to the inner ring of RAM-PnC and represented by din to excite the radial wave propagating from the inner ring to the outer ring, as shown in the figure. In order to eliminate the reflection effect of the outer boundary, the outer circle is surrounded by a perfectly matched layer (PML). It is assumed that there are good bonding conditions at the interface between the materials.

The transmission characteristics of the phononic crystal array obtained using the delay line model and the reference line (all materials are Si) are calculated in the frequency range of 0–300 MHz, as shown in Figure 5. The band gap is formed in the range of 122–225 MHz, and the propagation of the acoustic wave is adequately suppressed. As shown in Figure 5a, under the condition of two-dimensional simulation, the ability of acoustic waves to suppress reduces the acoustic wave transmission coefficient by 60 dB at a frequency of 175.14 MHz. As shown in Figure 5b, under the condition of three-dimensional simulation, the acoustic wave’s suppression ability reduces the acoustic wave transmission coefficient by 45 dB at a frequency of 175.14 MHz.

## 3. The Design of TPOS Resonator

Figure 6 shows the three-dimensional structure diagram of the resonator designed in this paper. The resonator structure is composed of three layers of materials. From the top to the bottom, the aluminum metal with a thickness of 0.5 m as the upper electrode, the AlN piezoelectric film with a thickness of 0.1 μm and the bottom Si structure layer with a thickness of 10 μm is used as the lower electrode of the resonator. Applying a voltage to the metal upper electrode generates a vertical electric field between the metal upper electrode and the silicon Si structure layer, as the lower electrode is on the piezoelectric film. Due to the inverse piezoelectric effect of the piezoelectric material, and according to the *d*_31_ piezoelectric coefficient of the AlN material, the electric field causes the plane expansion deformation of the piezoelectric film, thereby exciting the resonator to cause lateral expansion mode resonance.

The top view of the resonator designed in this paper is shown in Figure 7, which clearly shows the characteristics of the designed resonator with phononic crystals composed of radial alternating materials. A reflective structure is added to the support beam to constrain more energy in the resonator. The structure is divided into a multi-layer periodic structure with alternating high and low equivalent acoustic impedance. When the acoustic wave transmitted from the resonator passes through the reflection block, the acoustic wave is reflected at the interface between the periodic structural layers. Part of the energy transmitted to the reflection block is reflected by the resonator, which suppresses the energy loss of the resonator to a certain extent, thereby improving the Q value of the device. Figure 6b shows the schematic diagram of the reflective structure with alternating materials. The structural parameters of the designed resonator are marked in Figure 7. The detailed parameters are listed in Table 3.

The simulation model is shown in Figure 6. Since the overall structure of the resonator is symmetrical, only a quarter of the model can be established in order to reduce the amount of calculation during the simulation. The simulation of the complete model can be equivalent by assigning ‘symmetrical‘ boundary conditions to the symmetrical surface. In addition, a perfect matching layer needs to be set on the periphery of the resonator to absorb the elastic waves propagating outward to avoid the influence of elastic wave reflection on the simulation results. The width of the perfect matching layer is generally set to three times the wavelength so that it can fully absorb the elastic waves propagating out.

The above two resonators were modeled and simulated using the finite element simulation software COMSOL Multiphysics 5.4 to study the properties of their nine-order lateral expansion modes, as shown in Figure 6. The order of the resonator is set to nine. Therefore, the width of the resonator is 215.55, and the length is 3 times the width of 646.65. In the design of the device, the wavelength (i.e., *λ*) corresponding to the acoustic wave at the operating frequency of the resonator is designed to be twice the width of the adjacent interdigital electrode. Based on this, the following formula calculates the mechanical resonance frequency of the device:(10)$$f_{0}=\frac{v_{p}}{\lambda }=\frac{n}{2L}\sqrt{\frac{E}{\rho }}$$
where *v_p_* is the phase velocity associated with the lateral spreading mode, *E* and *ρ* denote Young’s modulus and the density of the Si layer, respectively, *n*. This provides the order of the resonant mode.

The mode of the resonator obtained via a simulation is shown in Figure 7. Figure 7a shows the total displacement distribution of the nine-order broadened resonance mode of the traditional resonator. Figure 7b shows the total displacement distribution of the ninth-order-stretched resonant mode of the RAM-PnC resonator. Figure 7c shows the *Z*-direction displacement distribution of the ninth-order-stretched resonant mode of the ordinary resonator. Figure 7d shows the *Z*-direction displacement distribution of the ninth-order-stretched resonant mode of the RAM-PnC resonator. The simulation results show that integrating the RAM-PnC array on the resonator can increase the resonator’s quality factor from 60,596 to 659,536,011.

In addition, by comparing the maximum vibration amplitude of the two resonators, it is shown that the integration of the RAM-PnC array on the resonator can concentrate more acoustic waves in the resonator, so the amplitude on the resonator is more significant, and the performance of the resonator is better. To quantitatively analyze the anchor loss of the substrate, the total displacement and *Z*-direction displacement at the A-A′ and B-B′ intercepts on the substrate are extracted, as shown in Figure 8. From Figure 8a–d, it can be seen that the integration of the RAM-PnC array on the resonator is more effective in suppressing the anchor loss. The maximum total displacement at the A-A′ cross-section is increased from 0.194 µm to 0.248 µm, and the maximum *Z*-direction displacement at the A-A′ cross-section is increased from 0.06 µm to 0.075 µm. The maximum total displacement at the B-B′ intercept is increased from 0.169 µm to 0.208 µm, and the maximum *Z*-direction displacement at the B-B′ intercept is increased from 0.017 µm to 0.0215 µm.

## 4. Discussion

To further obtain the equivalent circuit parameters of the resonator, the frequency domain simulation of the resonator based on 50-ohm impedance matching is carried out using the finite element method, and the Y11 curve and S21 curve are obtained, respectively. The simulation results are shown in Figure 9 and Figure 10. From Equations (11)–(14), the various parameter values of the piezoelectric MEMS resonator are solved. Since there is no load in the simulation process, the no-load quality factor (*Q_u_*) can be obtained by calculating the 3dB bandwidth.
(11)$$Q_{l}=\frac{f_{s}}{\Delta f_{-3dB}}$$
(12)$$Q_{u}=\frac{Q_{l}}{1-10^{-\frac{IL}{20}}}$$
(13)$$R_{m}=\frac{1}{\max\left\{Re\left(Y11\right)\right\}}$$
(14)$$K_{eff}^{2}=\frac{f_{p}^{2}-f_{s}^{2}}{f_{p}^{2}}$$
where Δ*f*_−3dB_ is the −3 dB bandwidth, *IL* is insertion loss, max {*Re*(*Y*11)} is the maximum real part of admittance, *f_p_* is the frequency at which the impedance amplitude is at the maximum, and *f_s_* is the frequency when the impedance amplitude is the minimum.

The Figure of merit (*FoM*) values between resonators can be compared to measure the performance of MEMS resonators. The definition of the piezoelectric resonator figure of *FoM* is proportional to *Q* and the coupling factor. The calculation formula is as follows:(15)$$FoM=k_{eff}^{2}\times Q$$

The above analysis and calculation compare the specific performance parameters of the traditional resonator and the RAM-PnC resonator at 175.14 MHz. The particular performance parameters obtained by the simulation in this paper are summarized in Table 4.

## 5. Conclusions

In this paper, a RAM-PnC structure is proposed. The finite element simulation method is used to calculate the dispersion curve and frequency response curve. The results show that RAM-PnC has a complete band gap width between 122 and 225 MHz, which can adequately isolate elastic waves. Then, the influence of different metal materials on the band gap is studied. By combining six kinds of metal materials (W, Al, Ag, Pt, Cu, Au) commonly used in MEMS with Si, it was found that the multi-material radial phononic crystal plate containing metal W produces the widest band gap. Generally, the wide band gap has better performance when preventing wave propagation, so the metal W is selected. Then, the optimal band gap structure is obtained by adjusting the width of the metal material W. Finally, RAM-PnC is implanted into the piezoelectric MEMS resonator, and the anchor quality factor is increased from 60,596 to 659,536,011. The dynamic impedance is reduced from 6.45 Ω to 0.08 Ω. The merit value (*FoM*) increases from 8.3 to 11.1.

## Figures and Tables

**Figure 1 micromachines-14-02241-f001:**
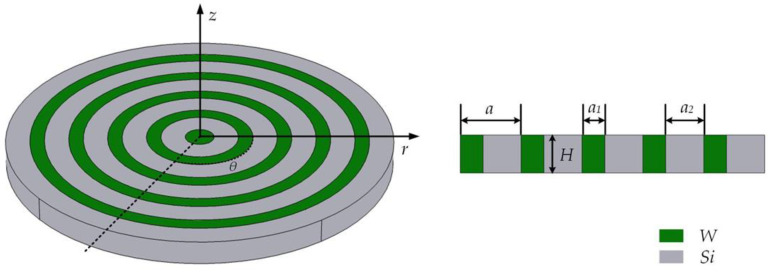
RAM-PnC three-dimensional schematic diagram and cross-section diagram.

**Figure 2 micromachines-14-02241-f002:**
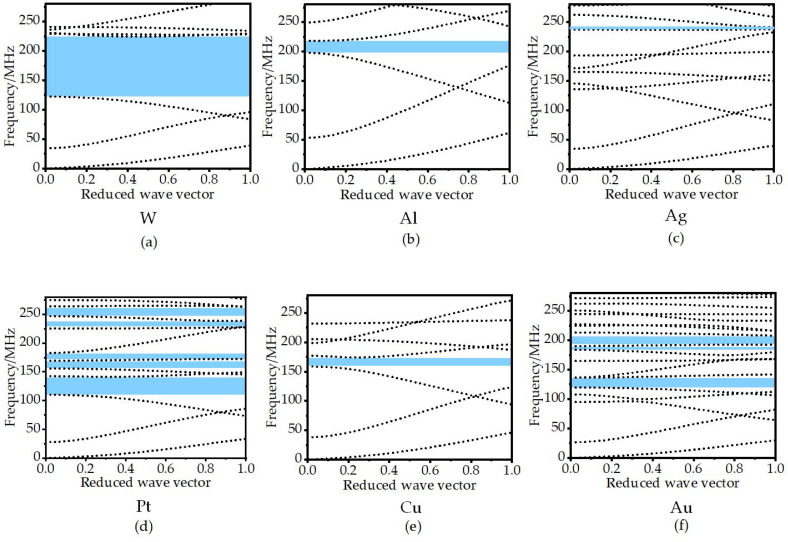
Dispersion curves formed by different metal materials combined with Si.

**Figure 3 micromachines-14-02241-f003:**
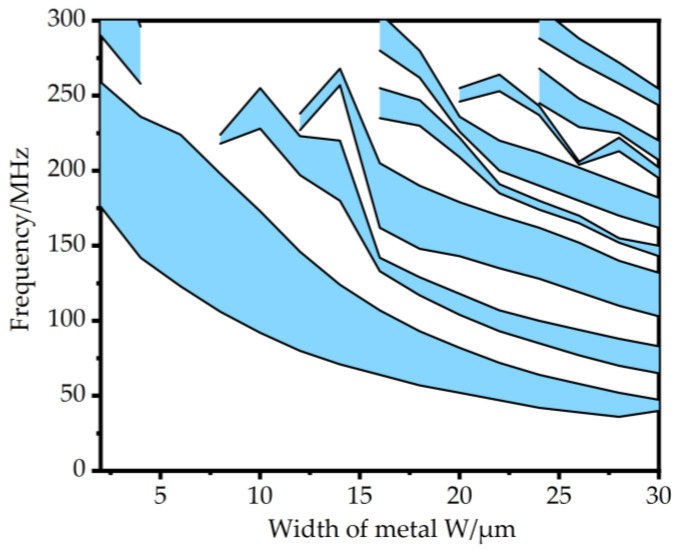
The dispersion curve formed by the combination of Si and metal W with different widths.

**Figure 4 micromachines-14-02241-f004:**
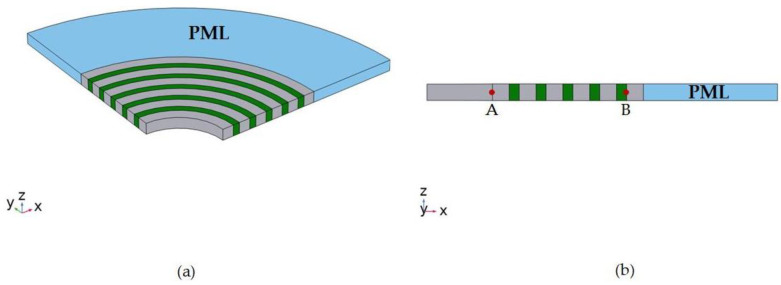
Five-period RAM-PnC transmission line model. (**a**) 3D schematic diagram and (**b**) 2D cross sectional diagram. Point A represents input (excitation signal) and point B represents output signal.

**Figure 5 micromachines-14-02241-f005:**
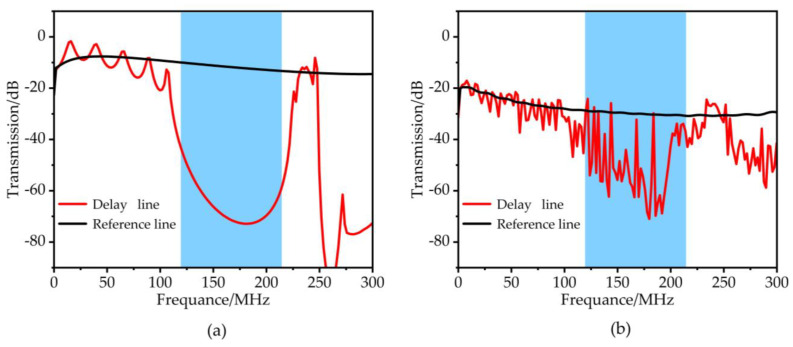
The simulation results of the RAM-PnC transmission curve (122–225 MHz) under the conditions of 2D (**a**) and 3D (**b**), respectively.

**Figure 6 micromachines-14-02241-f006:**
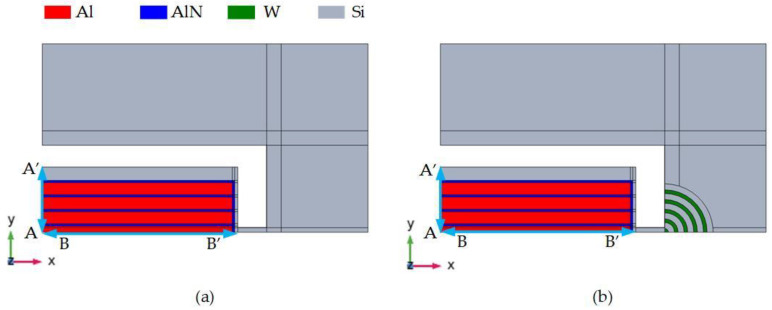
Simulation diagrams of quarter piezoelectric MEMS resonators: (**a**) the traditional-type TPOS MEMS resonator; (**b**) RAM-PnC TPOS MEMS resonator.

**Figure 7 micromachines-14-02241-f007:**
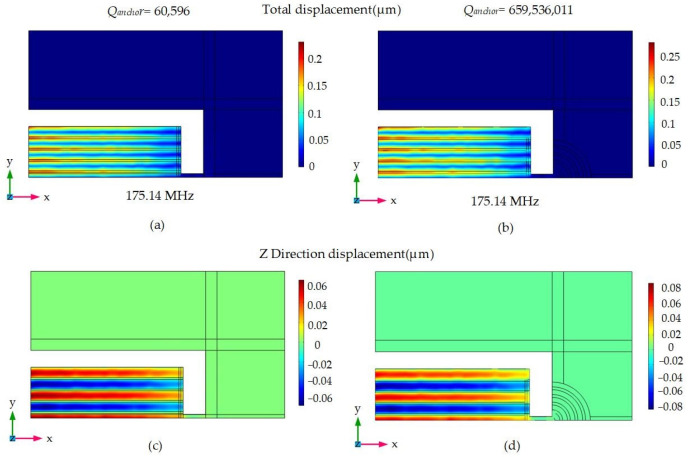
The total displacement distribution of the 9th-order width-extended resonant mode of (**a**) the ordinary resonator and (**b**) array plate resonator with RAM-PnC is shown in the figure; the Z-direction displacement distribution of the 9th-order width-extended resonant mode of the (**c**) ordinary resonator and (**d**) array plate resonator with RAM-PnC is shown in the figure.

**Figure 8 micromachines-14-02241-f008:**
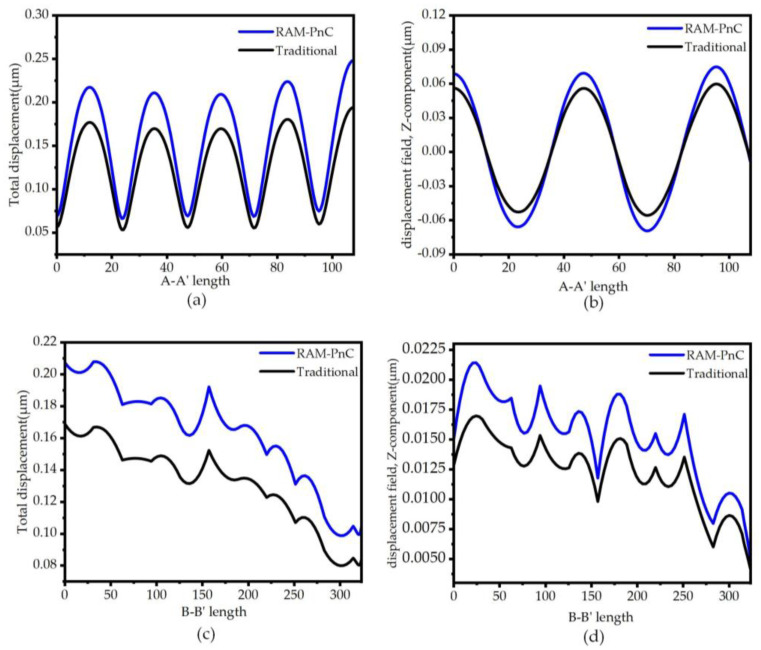
The (**a**) total displacement field (μm) diagram and (**b**) Z-direction displacement field (μm) diagram of the traditional resonator and the RAM-PnC resonator in the 9th-order width extension mode on the A-A′ cross-section, and the (**c**) total displacement field (μm) diagram and (**d**) *Z*-direction displacement field (μm) diagram of the traditional resonator and the RAM-PnC resonator in the 9th-order width extension mode on the B-B′ cross-section.

**Figure 9 micromachines-14-02241-f009:**
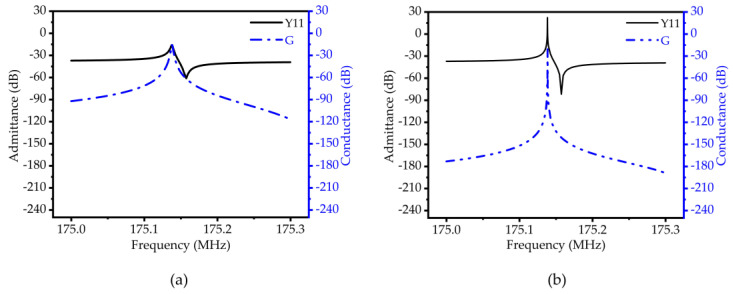
(**a**) Schematic diagram of admittance (Y11) and susceptance (G) curves of traditional TPOS MEMS resonators; (**b**) RAM-PnC MEMS resonator admittance (Y11) and susceptance (G) curve diagram.

**Figure 10 micromachines-14-02241-f010:**
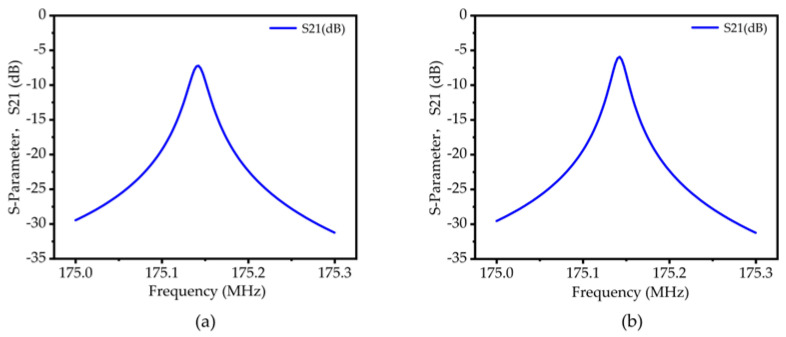
(**a**) Insertion loss (S21) curve of the traditional TPOS MEMS resonator; (**b**) Insertion loss (S21) curve of the RAM-PnC TPOS MEMS resonator.

**Table 1 micromachines-14-02241-t001:** The specific parameters of silicon set by the simulation in this paper.

Parameter Name (Abbreviated)	Value
Young’s modulus (*E*)	*E_x_* = *E_y_* = 169 GPa, *E_z_* = 130 GPa
Poisson’s ratio (*σ*) Shear modulus (*G*) Density (ρ)	*σ_xy_* = 0.064, *σ_yz_* = 0.36, *σ_zx_* = 0.28 *G_z_* = 50.9 GPa, *G_x_* = *G_y_* = 79.6 GPa 2330 kg/m^3^

**Table 2 micromachines-14-02241-t002:** Characteristic parameter table of different metals.

Materials	W	Al	Ag	Pt	Cu	Au
Density (g/cm^2^)	18.7	2.7	10.5	21.4	8.94	19.32
Longitudinal velocity (cm/s) × 10^2^	5.23	6.32	3.6	3.96	4.65	3.24
Acoustic impedance (g/cm^2^ s)	97.86	17.1	37.8	84.74	41.55	62.6

**Table 3 micromachines-14-02241-t003:** The specific size parameters of the resonators.

Parameters	Values (Unit)
Simulated resonant frequency (*f*_0_)	175 (MHz)
Wavelength (λ)	47.9 (µm)
Inter digitated transducer (IDT) finger (*n*)	9
Tethers width (*W_t_*)	15 (µm)
Tethers length (*L_t_*)	47.9 (µm)
Electrode gap (*G_e_*)	4 (µm)
Resonator width (*W_r_*)	215.55 (µm)
Resonator length (*L_r_*)	646.65 (µm)
Thickness of Al (*T_Al_*)	0.5 (µm)
Thickness of AlN (*T_AlN_*)	0.1 (µm)
Height of Si substrate (*HS*)	10 (µm)

**Table 4 micromachines-14-02241-t004:** Summary of the simulated result.

**Parameters**	**Traditional**	**RAM-PnC**
Resonant frequency (*f_r_*), MHz	175.14	175.14
Insertion loss (IL), dB	6.2	5.1
Motional resistance (*Rm*), Ω	6.45	0.08
Coupling coefficient (*K*^2^_*eff*_), %	0.0228	0.0228
*Q_anchor_*	60,596	659,536,011
Loaded quality factor (*Q_l_*)	8146	9467
Unloaded quality factor (*Q_u_*)	15,966	21,317
*FOM*	8.3	11.1

## Data Availability

Data are contained within the article.

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
