# Peer review of "Quality Factor Improvement of a Thin-Film Piezoelectric-on-Silicon Resonator Using a Radial Alternating Material Phononic Crystal"

_micromachines, 2023, doi:10.3390/mi14122241_

Round 1
Reviewer 1 Report
Comments and Suggestions for Authors
The paper is, in general, well written and very interesting, however there are one question and few editorial mistakes.
Very big admittance and Qanchor values compared with TOPS MEMS resonators are very hard to accept. Please check the conditions of the simulations. I am not saying that you are made a big mistake, but the values seem to me too high (even for theoretical calculation results).
Intead of body waves should be rather bulk waves used (e.g. lines 59, 60) for waves propagating in the volume of solid state.
If you use chemical symbols please do it consequently (like in W case). In the some lines Si symbol is used and in the other just name silicon.
Please correct the indexes x, y, z in the Table 1.
Please check inconsequent use of symbols. Inside the formulas the italic letters were used but inside the text normal ones (e. g. lines 115, 118, 119, 121, 196, ...). Please notice the difference betwin a and a.
Author Response
Thank you very much for review comments. Details
are shown in the attachment.

Reviewer 2 Report
Comments and Suggestions for Authors
The authors propose a RAM-PnC structure and apply it to the piezoelectric MEMS resonator. The issue is interested. There are some issues encouraged to be considered.
(1) To be consistent with the following, ra in Eq.8 needs to be changed to a.
(2) In Figure 2, different materials form different band gaps. What is the specific reason? Please give a brief explanation.
(3) What does the reference line in Figure 5 mean?
(4) Check the paper carefully and correct grammatical errors to improve the readability of the paper.
Comments on the Quality of English LanguageMinor editing of English language required
Author Response
Thank you very much for your review comments. Details are shown in the attachment.
